# Selective binding of the PHD6 finger of MLL4 to histone H4K16ac links MLL4 and MOF

Yi Zhang [1,7], Younghoon Jang [2,7], Ji-Eun Lee [2], JaeWoo Ahn[1], Longxia Xu[3], Michael R. Holden[1], Evan M. Cornett[3], Krzysztof Krajewski[4], Brianna J. Klein[1], Shu-Ping Wang[5], Yali Dou[6], Robert G. Roeder[5], Brian D. Strahl [4], Scott B. Rothbart [3], Xiaobing Shi [3], Kai Ge [2] & Tatiana G. Kutateladze [1]

Histone methyltransferase MLL4 is centrally involved in transcriptional regulation and is often mutated in human diseases, including cancer and developmental disorders. MLL4 contains a catalytic SET domain that mono-methylates histone H3K4 and seven PHD fingers of unclear function. Here, we identify the PHD6 finger of MLL4 (MLL4-PHD6) as a selective reader of the epigenetic modification H4K16ac. The solution NMR structure of MLL4-PHD6 in complex with a H4K16ac peptide along with binding and mutational analyses reveal unique mechanistic features underlying recognition of H4K16ac. Genomic studies show that one third of MLL4 chromatin binding sites overlap with H4K16ac-enriched regions in vivo and that MLL4 occupancy in a set of genomic targets depends on the acetyltransferase activity of MOF, a H4K16ac-specific acetyltransferase. The recognition of H4K16ac is conserved in the PHD7 finger of paralogous MLL3. Together, our findings reveal a previously uncharacterized acetyllysine reader and suggest that selective targeting of H4K16ac by MLL4 provides a direct functional link between MLL4, MOF and H4K16 acetylation.

---

[1] Department of Pharmacology, University of Colorado School of Medicine, Aurora, CO 80045, USA. [2] Laboratory of Endocrinology and Receptor Biology, National Institute of Diabetes and Digestive and Kidney Diseases, NIH, Bethesda, MD 20892, USA. [3] Center for Epigenetics, Van Andel Research Institute, Grand Rapids, MI 49503, USA. [4] Department of Biochemistry & Biophysics, The University of North Carolina School of Medicine, Chapel Hill, NC 27599, USA. [5] Laboratory of Biochemistry and Molecular Biology, The Rockefeller University, New York, NY 10065, USA. [6] Department of Pathology, University of Michigan, Ann Arbor, MI 48109, USA. [7]These authors contributed equally: Yi Zhang, Younghoon Jang. Correspondence and requests for materials should be addressed to K.G. (email: kai.ge@nih.gov) or to T.G.K. (email: tatiana.kutateladze@ucdenver.edu)

Human mixed lineage leukemia 4 (MLL4) or KMT2D is an evolutionarily conserved lysine methyltransferase that plays crucial roles in embryogenesis and development. MLL4 is frequently mutated in diseases, including Kabuki syndrome, congenital heart disorder and various cancers and is implicated in both tumor suppression and cancer cell proliferation[1]. Genomic studies have identified MLL4 as a major histone H3K4 mono-methyltransferase[2]. It catalyzes the transfer of a single methyl group from S-adenosyl-L-methionine (SAM) to the ε-amino group of Lys4, producing the epigenetic mark H3K4me1 primarily in transcriptional enhancer regions, where MLL4 co-localizes with lineage-determining transcription factors[3,4]. MLL4 is required for enhancer activation and cooperates with the H3K27/K18-specific acetyltransferases p300/CBP in the regulation of cell-type-specific transcriptional programs[4–6].

MLL4 belongs to a six-member MLL/KMT2 family of enzymes, with each member targeting H3K4 but being capable of generating distinct sets of H3K4 methylation states and at different genomic sites[1]. Much like other MLL/KMT2 family members, MLL4 functions in the context of a large multi-subunit complex consisting of the core subunits WDR5, Ash2L, RBBP5, and DPY30 and several accessory components, including NCOA6, PA1, PTIP, and UTX, with the latter being an H3K27 demethylase (Fig. 1a). MLL4 itself is a massive, ~600 kDa multimodular protein that harbors seven plant homeodomain (PHD) fingers, a pair of the FY-rich (FYRN and FYRC) domains, and the C-terminal catalytic Su(var)3–9, Enhancer of Zeste, Trithorax (SET) domain, also essential for protein stability in cells (Fig. 1b). With the exception of the SET domain, which is conserved in the MLL/KMT family, very little is known regarding functions of the other domains in MLL4[7]. The triple PHD finger cassette (PHD4–6) of MLL4 has recently been shown to associate with histone H4 either unmodified or asymmetrically dimethylated at R3 (H4R3me2a)[8].

Among the most common epigenetic modifications is acetylation of Lys16 in histone H4 (H4K16ac). This modification is necessary for a wide array of fundamental processes, including chromatin decompaction, DNA damage repair and RNA Polymerase II pause-release[9–11]. In mammals and flies, H4K16ac is generated or "written" by the histone acetyltransferase Males absent On the First (MOF), which is involved in transcriptional regulation, stem cell self-renewal and early embryonic development and is dysregulated in human diseases[12–16]. Despite the crucial role of MOF-dependent H4K16 acetylation in normal cell processes and disease, a specific reader of this mark has not yet been identified.

In this study, we identify the PHD6 finger of MLL4 (MLL4-PHD6) as histone reader that exhibits high selectivity toward H4K16ac. We describe the molecular and structural basis for the recognition of H4K16ac by MLL4-PHD6 and show that MLL4 co-localizes with H4K16ac in an MOF acetyltransferase activity-dependent manner in a set of MLL4 target genomic sites. Our study provides direct evidence of functional correlation between the abilities of MOF and MLL4 to write and read, respectively, the major histone modification H4K16ac.

## Results

### The PHD6 finger of MLL4 is a reader of H4K16ac.
The region containing three PHD fingers (PHD4–6) of MLL4 has been shown to associate with histone H4[8]. To define the role of MLL4-PHD6 in this interaction, we expressed it as [15]N-labeled protein and tested by nuclear magnetic resonance (NMR; Fig. 1c). Titration of H4 peptide (residues 1–9 of H4) into the MLL4-PHD6 sample caused no chemical shift perturbations (CSPs) in [1]H,[15]N heteronuclear single quantum coherence ([1]H,[15]N HSQC) spectra,

indicating that MLL4-PHD6 does not bind to the nine N-terminal residues of H4 tail. Gradual addition of the longer peptide (residues 1–23 of H4, H4$_{1–23}$), however, resulted in large CSPs, which suggested that the C-terminal part of the histone H4 tail is recognized by MLL4-PHD6. Indeed, peptides encompassing residues 12–23 or 8–25 of H4 induced CSPs similar in directions to CSPs induced by the longer H4$_{1–23}$ peptide, supporting the idea that the C-terminal region of H4 is the target sequence of MLL4-PHD6 (Fig. 1c and Supplementary Figs. 1 and 2).

The C-terminal region of the H4 tail contains several lysine residues (K12, K16, and K20) known to be posttranslationally modified, predominantly acetylated. We therefore explored whether MLL4-PHD6 is capable of sensing the acetylation states of these lysine residues. We assessed binding of MLL4-PHD6 to H4K12ac, H4K16ac, and H4K20ac peptides (all, residues 1–23 of H4) using NMR, fluorescence and microscale thermophoresis (MST) assays (Fig. 1d–g and Supplementary Figs. 3–5). An almost identical pattern of CSPs in the intermediate exchange regime on the NMR timescale was observed in [1]H,[15]N HSQC titration experiments with H4K12ac, H4K20ac or H4 peptides, indicating that acetylation of H4K12 or H4K20 does not augment the histone-binding activity of MLL4-PHD6 (Fig. 1c and Supplementary Fig. 4). By contrast, upon titration of the H4K16ac peptide, some resonances of MLL4-PHD6 exhibited more pronounced line broadening, indicative of tight binding (Fig. 1d). In agreement, binding affinity of MLL4-PHD6 to H4K16ac peptide was found to be 1.1 μM as measured by tryptophan fluorescence, whereas binding of this domain to H4, H4K12ac, and H4K20ac peptides was 12–28-fold weaker (Fig. 1e, f and Supplementary Fig. 5). The selectivity of MLL4-PHD6 toward H4K16ac was further corroborated by MST experiments. The MST assays yielded a dissociation constant ($K_d$) of 2.1 μM for the interaction of MLL4-PHD6 with the H4K16ac peptide, whereas association with the H4 peptide was eight-fold weaker ($K_d = 17$ μM) (Fig. 1g). Although acetylation of H4K16 substantially enhanced binding, we found that the residues surrounding K16 are also required: a free Kac amino acid failed to induce CSPs in MLL4-PHD6 even at high Kac concentration (Supplementary Fig. 1). We note that the $K_d$ value for the MLL4-PHD6:H4K16ac interaction was in the range of binding affinities exhibited by the majority of histone-binding modules[17–19], reinforcing the idea that MLL4-PHD6 is a previously unidentified reader of H4K16ac (Fig. 1h).

### The mechanism of H4K16ac recognition by MLL4-PHD6.
To elucidate the molecular basis for the interaction of MLL4-PHD6 with H4K16ac, we determined the solution structure of the MLL4-PHD6:H4K16ac complex by NMR spectroscopy. The structural ensemble revealed a central double-stranded anti-parallel β-sheet and two α-helices of MLL4-PHD6 (Supplementary Figs. 6–9 and Supplementary Table 1). Two zinc atoms were coordinated by the C4HC3 motif, characteristic of PHD fingers. Residues Gly14-Arg17 of the H4K16ac peptide adopted an extended conformation and formed a third β strand pairing with the existing β-sheet of MLL4-PHD6 in an antiparallel manner (Fig. 2a). Intermolecular nuclear Overhauser enhancements were identified between residues Gly14, Ala15, Lys16ac and Arg17 of H4 and residues L1519, L1520, I1521, Q1522, W1529, E1540, V1543 and A1547 of MLL4-PHD6. Overall, the PHD6 finger surface where the Gly13-Lys16ac sequence of H4K16ac is bound was largely hydrophobic (Fig. 2b). However, this hydrophobic site was surrounded by the highly negatively charged walls created by the side chains of residues in the loop preceding the β1 strand and the acidic α-helix of PHD6, including E1516, E1517, D1518, E1540, E1544, and D1548.

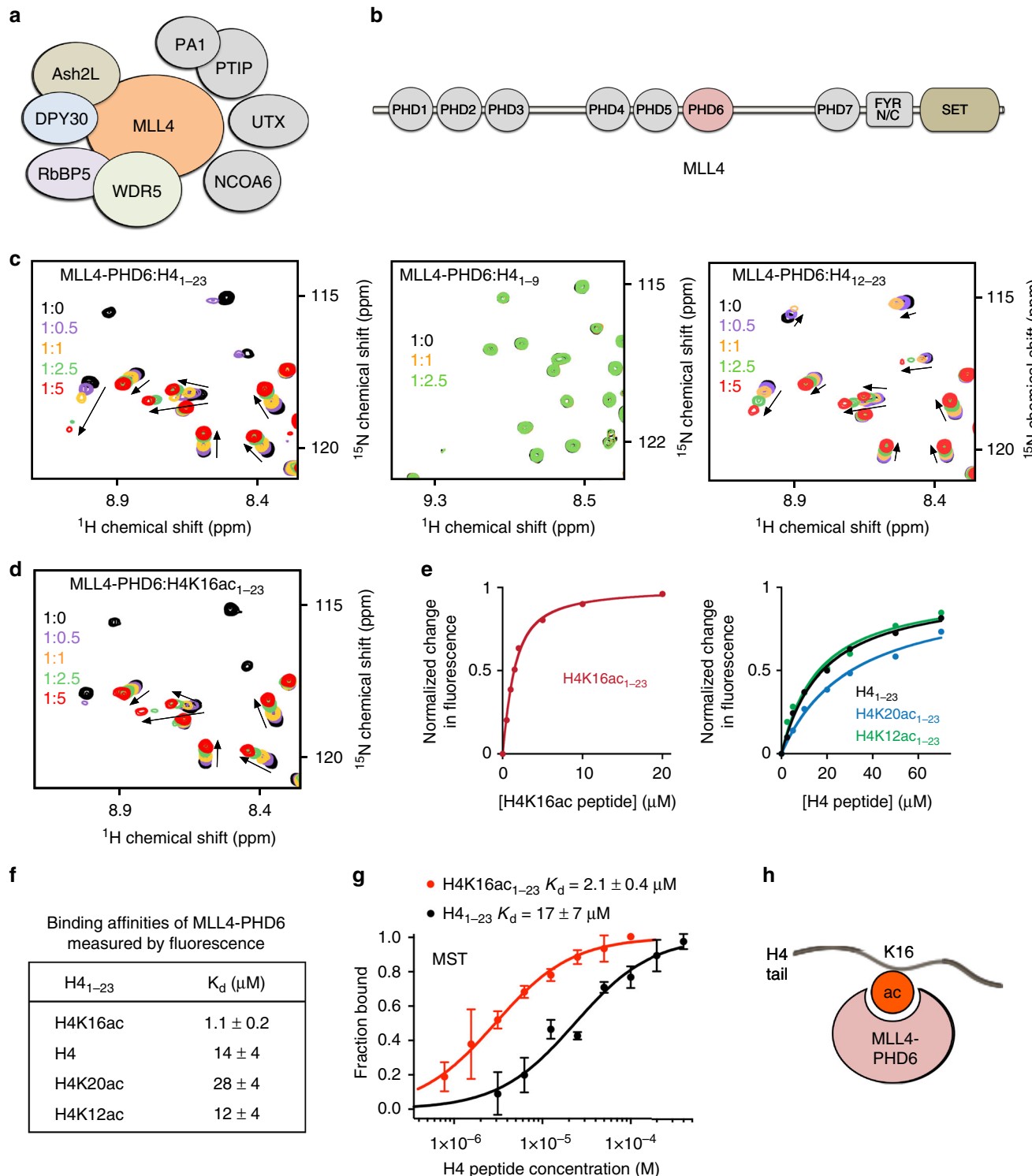

**Fig. 1** MLL4-PHD6 recognizes the histone mark H4K16ac. **a** Schematic of the MLL4 complex. **b** MLL4 domain architecture. **c** Superimposed $^1$H,$^{15}$N heteronuclear single quantum coherence (HSQC) spectra of MLL4-PHD6$_{1503-1562}$ collected upon titration with H4$_{1-23}$, H4$_{1-9}$, and H4$_{12-23}$ peptides. Spectra are color coded according to the protein:peptide molar ratio. See also Supplementary Fig. 1, first panel. **d** Superimposed $^1$H,$^{15}$N HSQC spectra of MLL4-PHD6$_{1503-1562}$ collected upon titration with H4K16ac$_{1-23}$ peptide. Spectra are color coded according to the protein:peptide molar ratio. **e** Representative binding curves used to determine the $K_d$ values by fluorescence spectroscopy (also see Supplementary Fig. 5). **f** Binding affinities of wild-type MLL4-PHD6 for the indicated histone peptides measured by tryptophan fluorescence. Error represents s.d. in triplicate measurements. Source data are provided as a Source Data file. **g** Binding curves used to determine the $K_d$ values by microscale thermophoresis. Error represents s.d. in triplicate measurements. **h** A schematic showing specific reading of the H4K16ac mark by MLL4-PHD6

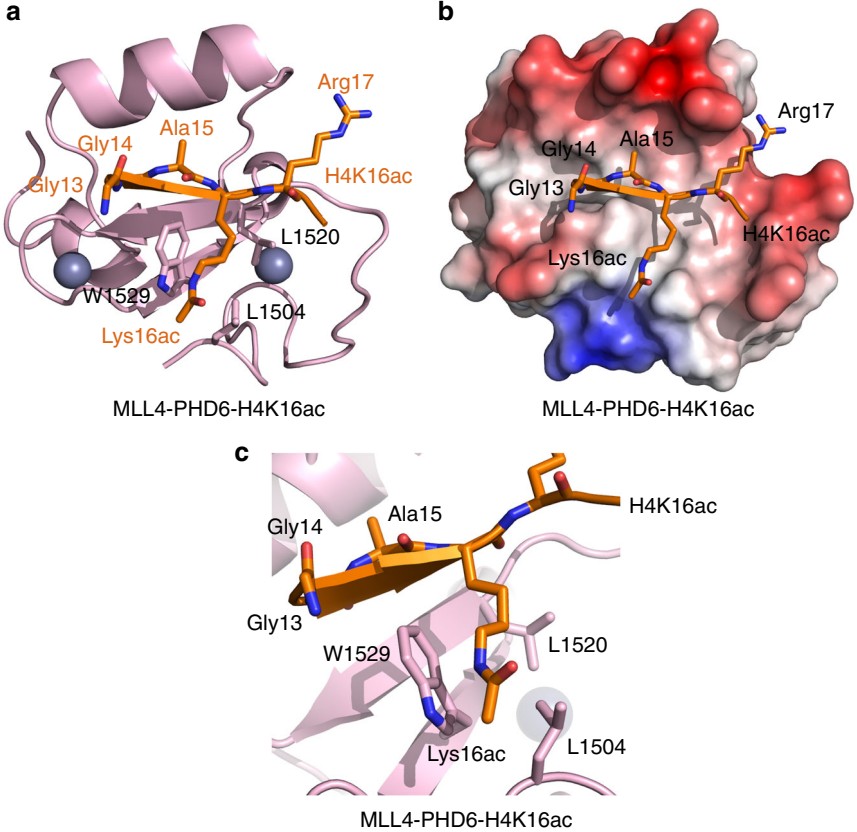

**Fig. 2** Structure of MLL4-PHD6 in complex with H4K16ac peptide. **a** A ribbon diagram of the solution nuclear magnetic resonance structure of MLL4-PHD6 (pink) in complex with the H4K16ac peptide (orange). **b** Electrostatic surface potential of MLL4-PHD6 colored blue and red for positive and negative charges, respectively. The bound H4K16ac peptide is shown in a stick model. **c** A zoom-in view of the H4K16ac-binding pocket

The structure of the complex has illuminated two unique mechanistic features underlying the recognition of H4K16ac. One is the hydrophobic groove formed by the residues L1504, L1520, and W1529 of MLL4-PHD6, which accommodates the non-charged acetylated side chain of Lys16 (Fig. 2b, c). The aromatic side chain of W1529 is positioned orthogonally to the protein surface and is differently perturbed by acetylated and non-acetylated K16 (Supplementary Fig. 10a). The second feature is the presence of a highly flexible Gly13-Gly14 sequence in H4 that ensures proper fitting of the N-terminal part of the H4K16ac peptide in the binding site. Although Gly13 of H4 was not restrained and showed structural flexibility, it formed a sharp turn in the majority of structures in the ensemble, allowing Gly14 to occupy the hydrophobic cavity of MLL4-PHD6 and avoid steric clashes (Supplementary Fig. 9). These structural features point to a previously uncharacterized mechanism for the recognition of histone PTMs by PHD fingers and distinguish MLL4-PHD6 from other histone H4 readers (see discussion below).

**The contribution of the binding site residues**. To determine the role of interfacial residues in the MLL4-PHD6:H4K16ac complex formation, we generated a chimeric construct containing residues 11–23 of H4 coupled to the residues 1503–1560 of MLL4-PHD6 through a hepta-glycine linker (Fig. 3a). The $^1$H,$^{15}$N HSQC spectrum of the linked H4-MLL4-PHD6 construct showed that the H4 region is tightly bound to the protein. The spectrum (colored black in Fig. 3a) did not overlay with the spectrum of an isolated MLL4-PHD6 finger in the apo state (blue); however, it overlaid well with the spectrum of MLL4-PHD6 bound to H4$_{1-23}$ peptide (dark yellow). These data suggest that the linked and

unlinked MLL4-PHD6:H4 complexes have similar structures. Mutation of either H4K12 or H4K23 to alanine in the linked H4-MLL4-PHD6 construct resulted in only small spectral changes, confirming that K12 and K23 of H4 are dispensable in this interaction. In contrast, mutation of K16 and R17 to alanine disrupted formation of the complex, underscoring the critical role of these residues in the binding.

Superimposition of $^1$H,$^{15}$N HSQC spectra of MLL4-PHD6 bound to H4K16ac$_{1-23}$ peptide (red) and H4$_{1-23}$ peptide (dark yellow) revealed that the majority of crosspeaks overlap, indicating that both peptides are bound in a similar manner (Fig. 3a and Supplementary Fig. 10b). As expected, most notable differences were observed in resonances of the residues located near the K16-binding-site-forming W1529, L1520, and L1504. Not only the side chain NHε of W1529 but also backbone amides of W1529, R1528, L1519, and V1505 were differently perturbed by H4K16ac and H4 (Supplementary Fig. 10). Although W1529 and L1520 comprise the hydrophobic core of the domain and are likely necessary for structural stability (W1529A is unfolded, Supplementary Fig. 11), L1504 is more solvent exposed and can be substituted with Glu. We rationalized that mutation of the hydrophobic leucine residue to the negatively charged glutamate would reduce binding of hydrophobic K16ac and increase binding of positively charged unmodified K16. Indeed, compared to wild-type (WT) MLL4-PHD6, the ability of L1504E mutant to interact with H4K16ac was reduced ($K_d = 3.5\ \mu M$), whereas its ability to interact with non-acetylated H4 was enhanced ($K_d = 4.1\ \mu M$) (Fig. 3c and Supplementary Fig. 5). We concluded that hydrophobic nature of the Leu-Leu-Trp binding groove is a driving force in preference of MLL4-PHD6 for the hydrophobic side chain of K16ac.

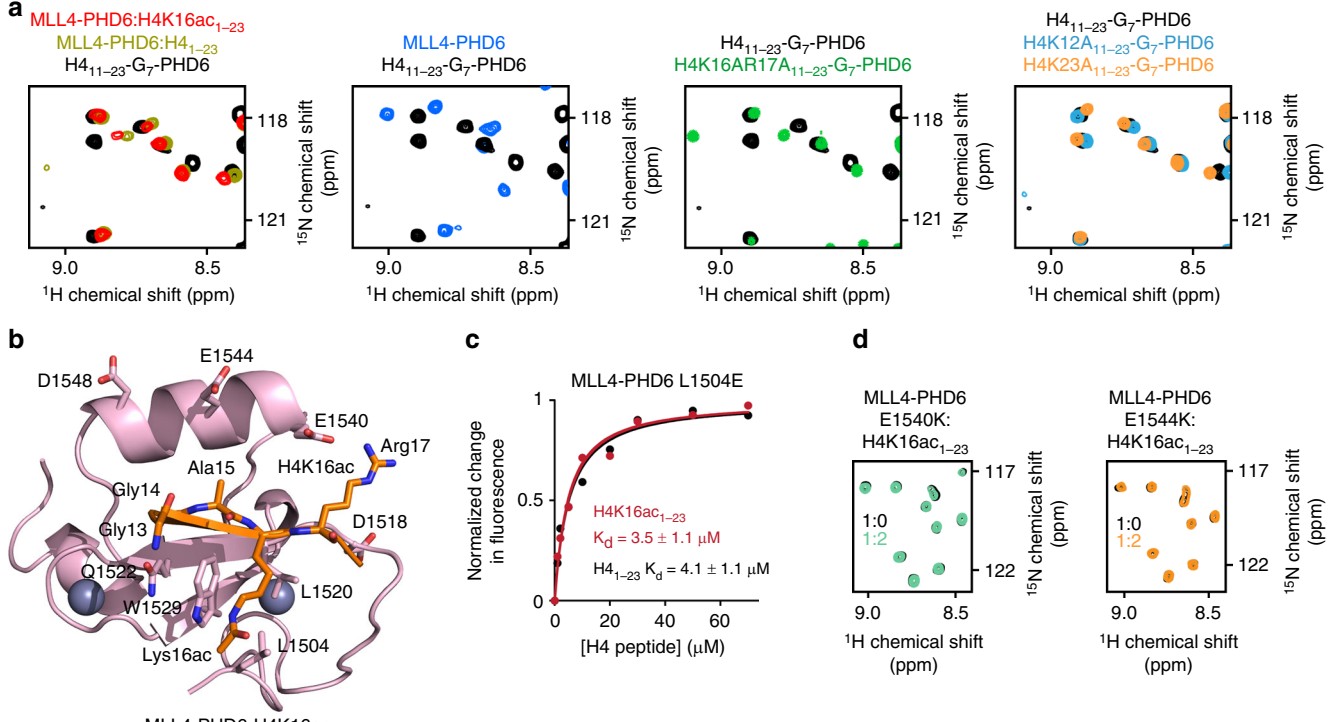

**Fig. 3** Molecular basis for the specific targeting of H4K16ac by MLL4-PHD6. **a** Superimposed $^1H,^{15}N$ heteronuclear single quantum coherence (HSQC) spectra of the linked $H4_{11-23}$-$G_7$-PHD6 construct, wild-type (black) and indicated mutants (green, light blue, and orange), the isolated MLL4-PHD6 domain (blue), isolated H4-bound MLL4-PHD6 (dark yellow), and isolated H4K16ac-bound MLL4-PHD6 (red). **b** The structure of the H4K16ac-bound MLL4-PHD6 finger. **c** Representative binding curves used to determine the $K_d$ values for the L1504E mutant by fluorescence spectroscopy (also see Supplementary Fig. 5). Source data are provided as a Source Data file. **d** Superimposed $^1H,^{15}N$ HSQC spectra of MLL4-PHD6 mutants collected upon titration with H4K16ac peptide. Spectra are color coded according to the protein:peptide molar ratio

Arg17 of the H4K16ac peptide is bound in a channel formed by two negatively charged clusters of MLL4-PHD6, $EED_{1518}$, and $EDDVE_{1544}$ (Fig. 3b). To assess the importance of these clusters in the interaction with H4K16ac, we generated D1518A, E1540K, and E1544K mutants and tested them by NMR. A lack of CSPs in $^1H,^{15}N$ HSQC spectra of E1540K and E1544K upon addition of H4K16ac peptide indicated that both glutamates contribute substantially to electrostatic interactions, especially E1540 that is in close contact with Arg17 of the peptide, and both are required for the binding (Fig. 3d). Mutation of D1518, as well as Q1522 and D1548, led to protein unfolding, implying that these residues are necessary for structural stability of MLL4-PHD6 (Supplementary Fig. 11).

**MLL4 colocalizes with H4K16ac in cells.** MLL4 has been shown to be essential for cell differentiation, including adipogenesis[2], therefore we employed a mouse preadipocytes model system to examine the functional significance of the H4K16ac recognition by MLL4. Because H4K16ac is generated by the MOF acetyltransferase, we isolated brown preadipocytes from $Mof^{f/f};Cre-ER$ mice in order to manipulate H4K16ac levels[10]. In this mouse strain, exons 4–6 of the $Mof$ gene are flanked by loxP sites (Fig. 4a). In the presence of 4-hydroxytamoxifen (4OHT), Cre-ER recombinase was able to delete exons 4–6 of the $Mof$ locus. The efficient deletion of the loxP-flanked $Mof$ exons in the preadipocytes was verified at the mRNA and protein levels. After two rounds of 4OHT treatment, the mRNA level of $Mof$ was significantly decreased, whereas mRNA of a control gene $Jmjd3$ was unaffected (Fig. 4b). Furthermore, the presence of truncated $Mof$ mRNA in the $Mof^{f/f};Cre-ER$ cells confirmed a deletion of $Mof$ (Fig. 4c). Western blot analysis also showed reduced MOF protein

levels, whereas the MLL4 complex remained intact (Fig. 4d). Importantly, the MOF knockout (KO) led to a substantial decrease in H4K16ac levels but had no effect on other histone marks, such as H3K4me1, H3K4me3, and H4K8ac (Fig. 4e). Consistent with the previous findings[10], $Mof$ KO also decreased cell growth in preadipocytes (Supplementary Fig. 12).

To determine whether the deletion of $Mof$ affects genomic occupancy of MLL4, we performed chromatin immunoprecipitation experiments followed by high-throughput sequencing (ChIP-Seq) in the $Mof^{f/f};Cre-ER$ preadipocytes treated with and without 4OHT. Both MLL4 chromatin association and H4K16ac levels were significantly reduced in the $Mof$ KO cells (Fig. 4f). A total of 52,820 MLL4-binding sites were identified from the control (Mock) or $Mof$ KO (4OHT) cells using the SICER method[20]. Importantly, these sites also had high level of H3K4me1, which is generated by MLL4[2]. Gene ontology (GO) data showed that MLL4 target genes are involved in a broad range of cellular events, including cell adhesion, metabolism, and mitosis (Supplementary Fig. 13). Further analysis of the 49,597 MLL4 target regions and the H4K16ac mark revealed that 17,457 (~35%) MLL4-binding sites overlapped with H4K16ac-enriched (H4K16ac⁺) sites in the control cells (Fig. 4g, h). In this subset of MLL4 target regions, MOF KO led to a concurrent decrease in H4K16ac levels and MLL4 association with chromatin, which is indicative of a direct correlation between MOF, MLL4, and H4K16 acetylation.

**MOF is required for binding of MLL4 to a subset of genes.** The MOF-dependent binding of MLL4 to specific genomic sites was characterized by comparing normalized MLL4 ChIP read counts between WT and $Mof$ KO cells. $Mof$ KO selectively inhibited

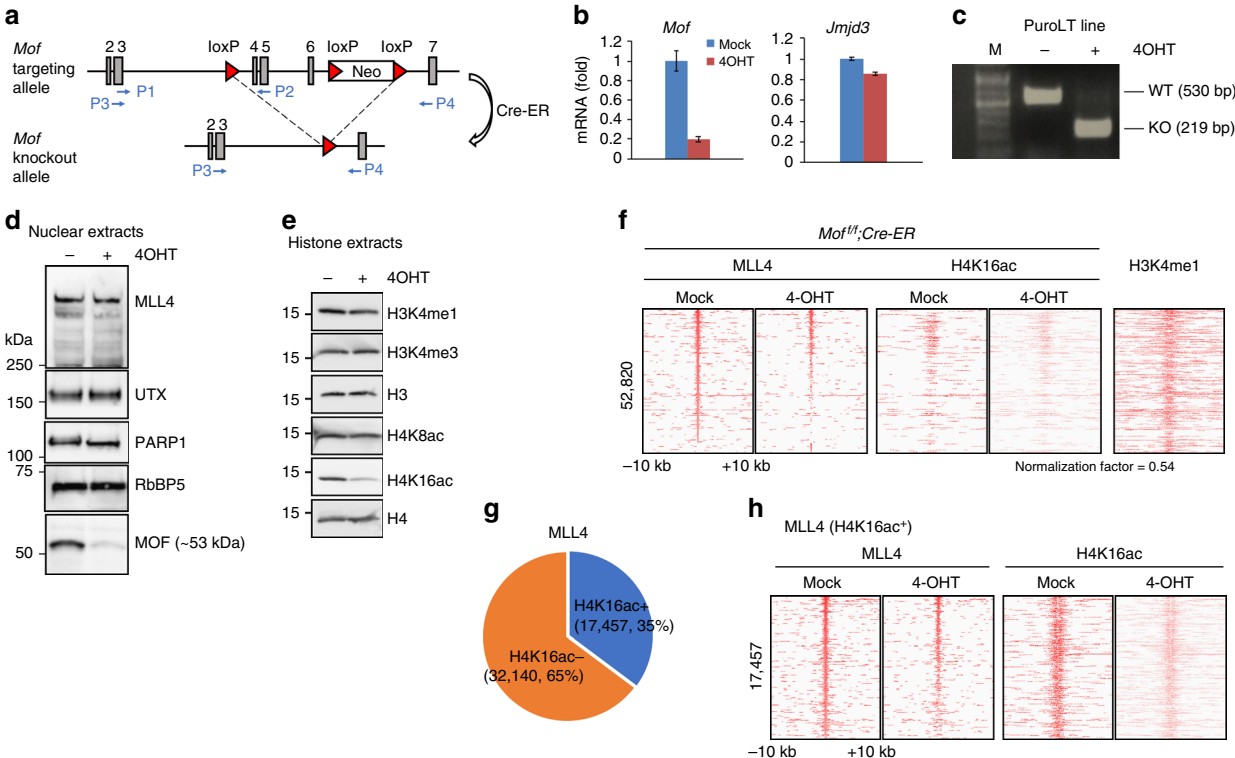

**Fig. 4** A subset of MLL4-bound genomic regions overlaps with H4K16ac. SV40T-immortalized $Mof^{f/f}$;Cre-ER brown preadipocytes were treated with 4-hydroxytamoxifen to delete Mof. Cells were collected for reverse transcriptase PCR (RT-PCR), western blot and chromatin immunoprecipitation–sequencing (ChIP-Seq) analysis. **a** Schematics of targeting allele and knockout allele of $Mof^{f/f}$ mice. In the targeting allele, a single loxP site was inserted in the intron before exon 4 and a neomycin selection cassette flanked by loxP sites was inserted in the intron after exon 6. The locations of RT-PCR primers P1–P2 (used in Fig. 4b) and P3–P4 (used in Fig. 4c) are indicated by arrows. **b** Quantitative RT-PCR confirmation of Mof deletion in preadipocytes. The data are presented as mean ± s.d. Two technical replicates from a single experiment were performed. Source data are provided as a Source Data file. **c** RT-PCR confirmation of wild-type and truncated Mof mRNA. **d** Western blot analysis of MLL4, UTX, and MOF. PARP1 and RbBP5 were used as loading controls. **e** Western blot analysis of histone modifications (also see Supplementary Fig. 15). **f** Heat maps around MLL4-binding sites. H4K16ac ChIP-Seq data were normalized with global H4K16ac levels measured by western blot. Published H3K4me1 data were obtained from GEO database (GSE74189)[49]. **g** Pie chart illustrating that 35% of MLL4-binding sites overlap with H4K16ac. **h** Heat maps of MLL4 and H4K16ac around 17,457 H4K16ac+ MLL4-binding sites

MLL4 binding to promoters of *Plin3*, the gene encoding a lipid droplet protein, and *Zhx1*, the gene encoding a zinc-finger and homeodomain transcription factor, but not to promoters of nearby genes (Fig. 5a). Overall, the deletion of *Mof* affected genomic occupancy of MLL4 on all sites (Fig. 5b). Interestingly, the H4K16ac+ MLL4-binding sites were located almost exclusively on active enhancers and active promoters, which is consistent with the previous reports that H4K16ac is enriched around transcription start sites and enhancers of active genes in embryonic stem cells[21] (Fig. 5c). While ~30% of MLL4-binding sites on active enhancers overlapped with H4K16ac, over 82% of MLL4-binding sites on active promoters were H4K16ac positive. Within 17,457 H4K16ac+ MLL4-binding regions, the deletion of *Mof* resulted in the loss of 4591 (~26%) MLL4-binding sites (Fig. 5d). GO analysis revealed that these 4591 *Mof*-dependent MLL4 target regions are associated with the genes that define mesenchymal lineages (bone marrow development, platelet formation, and adipose tissue development) or specific brown adipose lineages (positive regulation of mitochondrion organization) (Fig. 5e). The 4591 *Mof*-dependent MLL4-binding regions were enriched in motifs of AP-1 and ETS families of transcription factors (Fig. 5f).

**MOF catalytic activity-dependent binding of MLL4 to chromatin.** To substantiate the biological significance of MOF-mediated H4K16ac recognition by MLL4, we ectopically

expressed WT or an enzymatically dead mutant (K274R) of rat MOF (rMOF) in $Mof^{f/f}$;Cre-ER preadipocytes and deleted the endogenous *Mof* by treating the cells with 4OHT (Fig. 6a). In total, we identified 5296 MLL4-binding sites that were MOF enzymatic activity dependent (Fig. 6b). MLL4 binding to these 5296 regions was fully rescued by ectopic expression of WT MOF but not by ectopic expression of the catalytically dead MOF K274R mutant. MOF enzymatic activity-dependent MLL4 binding and H4K16ac enrichment was selectively observed in the proximal promoter regions of the *Itf81*, *Ccnd3*, and *Zfp13* genes (Fig. 6c–e). GO analysis indicated that MOF enzymatic activity-dependent MLL4-binding regions are associated with the genes that are necessary to block fat cell differentiation and maintain preadipocyte status, in addition to the genes regulating general biological functions (Fig. 6f). Motif analysis revealed that MOF enzymatic activity-dependent MLL4-binding sites were strongly enriched with AP-1 transcription factor motifs (Fig. 6g).

**H4K16ac recognition is conserved in the PHD7 finger of MLL3.** Much like MLL4 that contains seven PHD fingers, another member of the MLL/KMTD family, MLL3, harbors eight PHD fingers (Fig. 7). Alignment of sequences for all PHD fingers from human MLL4 and MLL3 suggests that MLL4-PHD6 has a high degree sequence similarity to MLL3-PHD7. We tested whether H4K16ac-binding activity is conserved in MLL3-PHD7 by NMR.

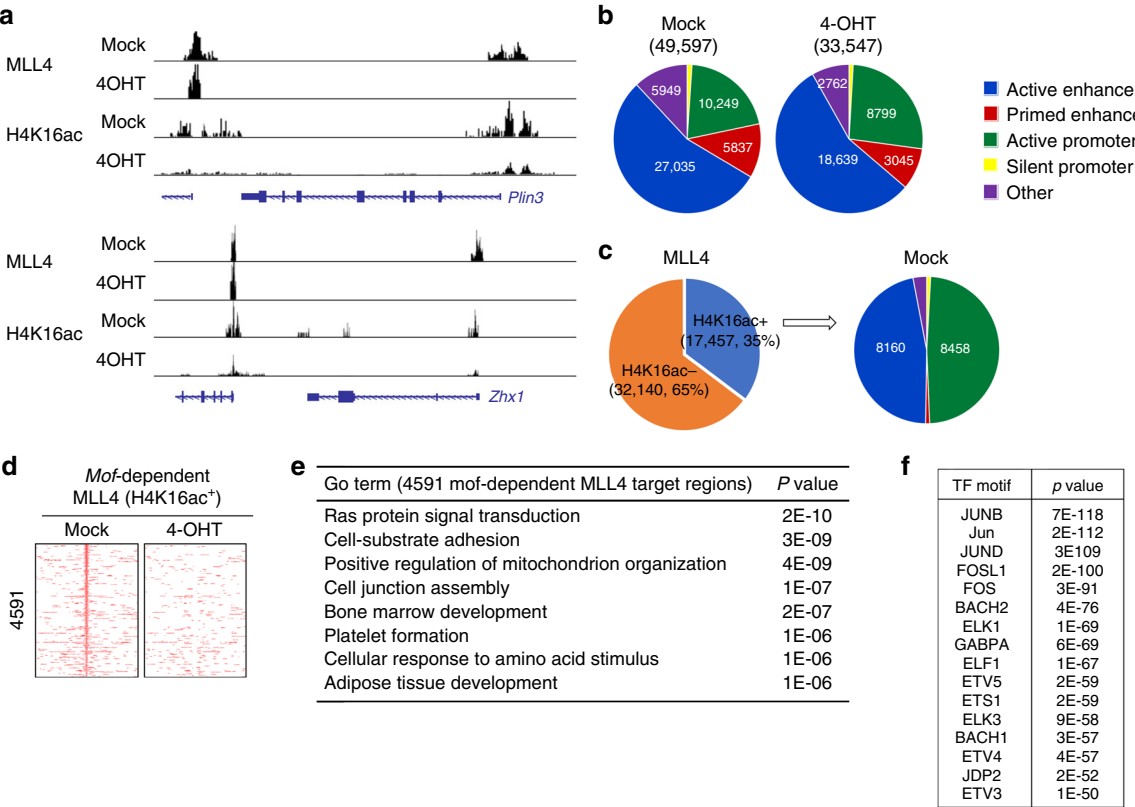

**Fig. 5** Targeting of MLL4 to active promoters is dependent on H4K16ac mark. *Mof*f/f;*Cre-ER* brown preadipocytes were treated with 4-hydroxytamoxifen and collected for chromatin immunoprecipitation–sequencing (ChIP-Seq) analysis. **a** ChIP-Seq profiles of MLL4 and H4K16ac on *Plin3* and *Zhx1* gene loci. **b** Deletion of *Mof* decreases MLL4 binding to all sites. **c** H4K16ac-enriched (H4K16ac+) MLL4-binding regions are mainly located on active enhancers and active promoters. **d** Heat maps of 4591 *Mof*-dependent H4K16ac+ MLL4-binding sites. **e** Gene ontology analysis of the genes associated with 4591 *Mof*-dependent H4K16ac+ MLL4-binding regions identified in **d**. **f** Motif analysis of 4591 *Mof*-dependent H4K16ac+ MLL4-binding regions identified in **d**

Gradual addition of H4K16ac peptide to the [15]N-labeled MLL3-PHD7 finger led to notable CSPs, thus confirming that, similar to MLL-PHD6, the PHD7 finger of MLL3 is also capable of targeting H4K16ac (Fig. 7a).

## Discussion
The finding that MLL4-PHD6 and MLL3-PHD7 recognize H4K16ac has expanded the family of acetyllysine readers, which currently includes bromodomain, YEATS, and DPF, and a set of domains in certain proteins, such as a ZZ domain, double bromodomain, and a double PH domain that display selectivity toward acetylated lysine residues[22–28]. Structural comparison of the MLL4-PHD6:H4K16ac complex with acetyllysine-bound BD, YEATS, and DPF reveals distinctly different mechanisms underlying the recognition of acetyllysine. Not only the folds and acetyllysine-binding pockets of these readers are unlike, but the acetylated lysine residues are restrained through different sets of hydrogen bonding, π–π–π, and hydrophobic interactions (Fig. 7b–d). One similarity perhaps is the overall hydrophobic character of the acetyllysine-binding sites and the presence of aromatic residues that likely facilitate engagement with the hydrophobic side chain of acetyllysine.

The PHD finger has been well established as a reader of the histone H3 tail, either trimethylated at Lys4 (H3K4me3) or unmodified[29–34], and the finding of a subset of PHD fingers that select for acetylated H4K16 is unexpected. Comparison of the MLL4-PHD6:H4K16ac structure with the ING2-PHD:H3K4me3 structure reveals some similarities in the histone-binding modes (Fig. 7e, f). In both complexes, histone tails are bound in an extended conformation and K16ac of H4 and K4me3

of H3 occupy binding sites made of aromatic and hydrophobic residues (Fig. 7f). The W1529 residue in the MLL4-PHD6:H4K16ac complex that forms one of the walls in the acetyllysine-binding groove is located in a similar position as an invariable Trp residue that forms a wall in the trimethyllysine-binding cage (W238 in ING2) (Fig. 7f, g). However, the aromatic cage for trimethylated lysine is larger in size and contains at least two aromatic residues that are involved in cation–π interactions with the trimethylammonium group of K4me3 and are strictly required for the recognition of H3K4me3. In contrast, K16ac lies in a narrow, single aromatic residue-containing groove with the acetyl moiety being placed between the side chains of tryptophan and leucine residues. A critical role of the free amino-terminal Ala1 residue in the recognition of either H3K4me3 or unmodified H3 is well established, but the MLL4 PHD6 finger binds to the middle part of histone H4 tail and does not require the presence of NH$_3^+$. Instead, a highly flexible Gly13-Gly14 sequence of H4 allows for the proper fitting of H4K16ac in the binding pocket of MLL4-PHD6. Alignment of amino acid sequences of seven PHD fingers from MLL4 and eight PHD fingers from MLL3 shows that the H4K16ac-binding site residues of MLL4-PHD6 are conserved in MLL3-PHD7 but not in other PHD fingers of MLL4 and MLL3 for which biological roles remain unidentified (Fig. 7g).

Our results demonstrate that MLL4 co-localizes with H4K16ac in vivo and that the association of MLL4 with a set of genomic targets is *Mof* dependent and requires the acetyltransferase activity of MOF. These data imply a hitherto unrecognized pathway for chromatin regulation that is likely central to the control of gene transcription. We propose that MOF-mediated deposition of H4K16ac at certain genomic sites directs the

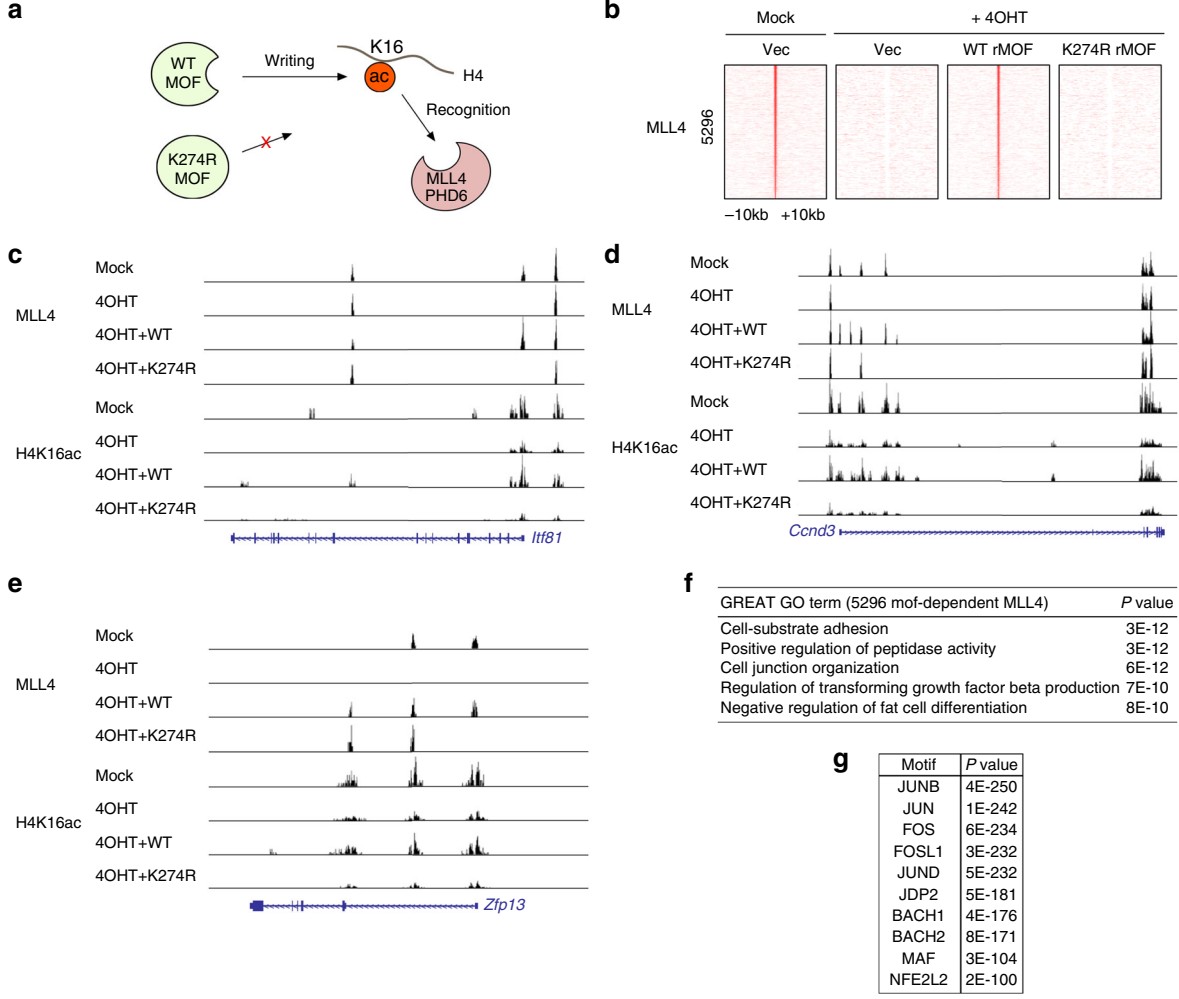

**Fig. 6** MOF (Males absent On the First) catalytic activity is necessary for *Mof*-dependent chromatin targeting of MLL4. *Mof*^f/f;*Cre-ER* brown preadipocytes were infected with lentiviruses expressing wild-type (WT) or enzyme dead mutant (K274R) form of rMOF, followed by 4-hydroxytamoxifen treatment. Cells were collected for chromatin immunoprecipitation–sequencing (ChIP-Seq) analysis of MLL4 and H4K16ac. **a** Schematic representation of WT and enzyme dead MOF. **b** Heat maps of 5296 MOF enzyme activity-dependent MLL4-binding sites. **c–e** ChIP-Seq profiles of MLL4 and H4K16ac on *Zfp13* (**c**), *Ccnd3* (**d**), and *Itf81* (**e**) gene loci. **f** Gene ontology analysis of the genes associated with 5296 MOF catalytic activity-dependent MLL4-binding regions. **g** AP-1 family transcription factor motifs are enriched at MOF catalytic activity-dependent MLL4-binding regions. Top 3000 MLL4-binding regions were used for motif analysis

recruitment or stabilization of MLL4 via its PHD6 finger recognition of H4K16ac. Furthermore, we show that MOF enzymatic activity-dependent MLL4-binding sites are highly enriched in the AP-1 transcription factor motif, therefore AP-1 could cooperate with MOF and H4K16ac in the recruitment of MLL4 to active transcriptional regions. Future studies will be needed to determine the broad significance of this mechanism in gene transcription and whether it is conserved across other eukaryotes.

## Methods

**Plasmids, antibodies, and chemicals**. The lentiviral pLenti-CMV plasmids expressing MYC-tagged WT or K274R mutant of rat MOF (a gift from Jianrong Lu) were described previously[35]. The following homemade antibodies have been described: anti-MLL4#3[36] and anti-UTX[37]. Anti-MOF (A300–992A) and anti-RbBP5 (A300–109A) were from Bethyl Laboratories. Anti-H3 (ab1791), anti-H3K4me1 (ab8895), and anti-H4 (ab7311) were from Abcam. Anti-H3K4me3 (07–473), anti-H4K8ac (07–328), and anti-H4K16ac (07–329) were from Millipore. Anti-PARP1 (556362) was from BD Bioscience. Anti-MYC (sc40) was from Santa Cruz and 4OHT (H7904) was from Sigma. A 1:1000 dilution of antibodies was used for western blot.

**Mouse strains and immortalization of primary brown preadipocytes**. *Mof*^f/f mice were crossed with *Cre-ER* (Jackson no. 008463) to generate *Mof*^f/f;*Cre-ER*

mice[10]. Primary brown preadipocytes were isolated from interscapular brown adipose tissue of newborn (P0) *Mof*^f/f;*Cre-ER* pups and immortalized by SV40T[38]. All mouse experiments were performed in accordance with the NIH Guide for the Care and Use of Laboratory Animals and approved by the Animal Care and Use Committee of NIDDK, NIH. For rescue experiment by ectopic expression of rat MOF, immortalized *Mof*^f/f;*Cre-ER* preadipocytes stably expressing WT or K274R MOF were established using lentivirus-mediated gene transfer. Briefly, 293FT lentiviral packaging cells were transfected with pLenti-CMV plasmids expressing MYC-tagged WT or K274R rat MOF. Preadipocytes were infected with those lentivirus-containing media, followed by puromycin selection for two passages. Those expressing MYC-tagged rat MOF were confirmed by western blot.

**Protein expression and purification**. The PHD6 domain of human MLL4 (aa 1503–1562) was cloned into a pCool vector. The H4-G7-MLL4-PHD6 construct (aa 11–23 of H4, hepta-glycine (GGGGGGG) linker, aa 1503–1562 of MLL4-PHD6) was cloned into a pDEST-15 vector. The human MLL3 PHD7 domain (aa 1083–1143) was cloned into a pGEX-6p vector. All constructs were expressed in Rosetta2 (DE3) pLysS cells. Protein production was induced with 0.5 mM IPTG overnight at 16 °C in Luria broth (LB) or minimal media (M9) supplemented with $^{15}NH_4Cl$ and 0.06 mM $ZnCl_2$ or $^{15}NH_4Cl$, $^{13}C$-glucose, and 0.06 mM $ZnCl_2$. The GST-tagged PHD proteins were purified on glutathione Sepharose 4B beads (GE Healthcare) in 50 mM Tris-HCl (pH 7.5) buffer, supplemented with 250 mM NaCl and 2 mM dithiothreitol (DTT). The GST tag was cleaved with Thrombin or TEV protease overnight at 4 °C. Unlabeled proteins were further purified by size exclusion chromatography and concentrated in Millipore concentrators. All

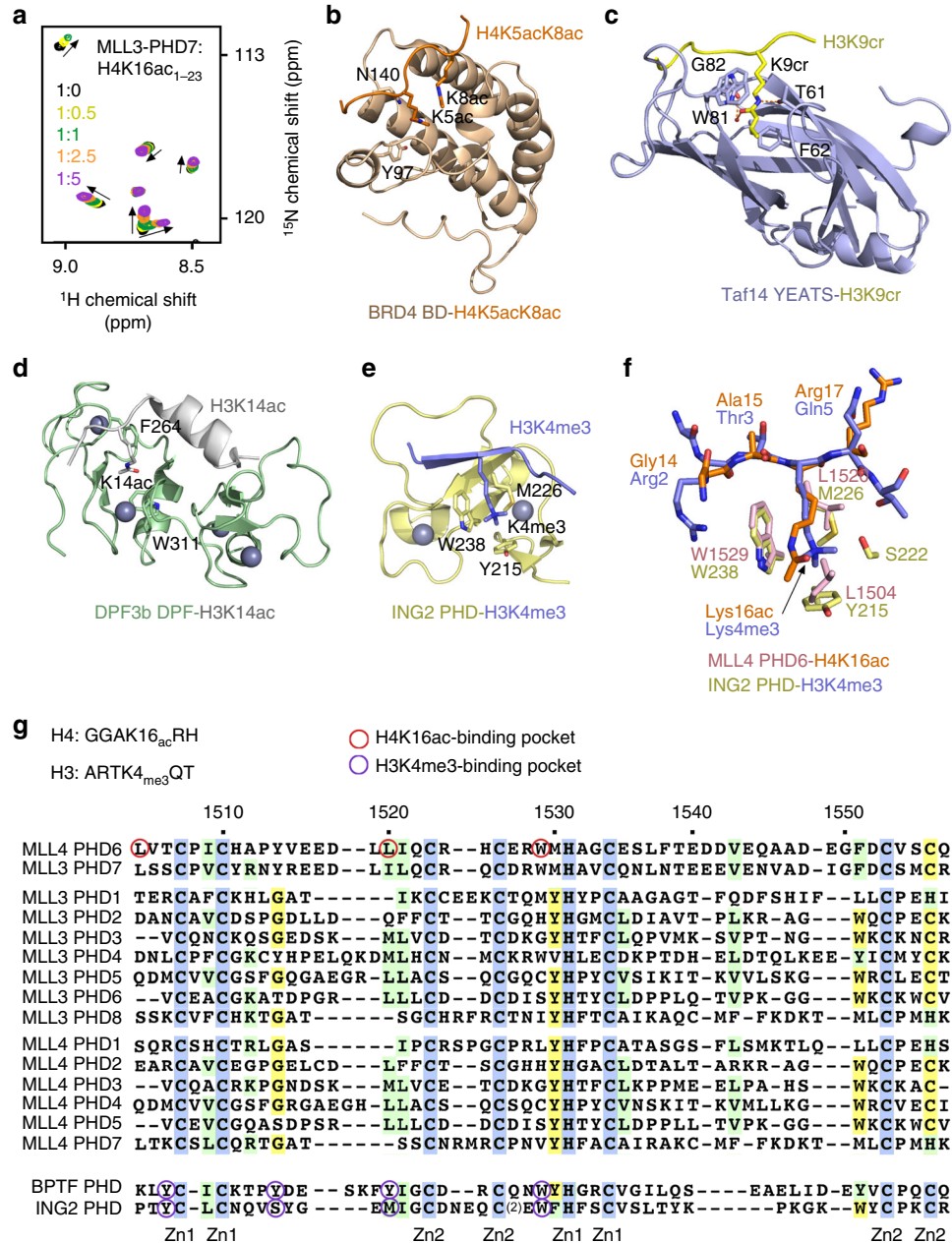

**Fig. 7** Comparative analysis of the H3K4me3 and H4K16ac recognition by histone readers. **a** Superimposed $^1$H,$^{15}$N heteronuclear single quantum coherence spectra of MLL3-PHD7$_{1083-1143}$ collected upon titration with H4K16ac peptide. Spectra are color coded according to the protein:peptide molar ratio. **b–d** Structures of the BRD4 BD, Taf14 YEATS, and DPF3b DPF readers in complex with the indicated histone peptides[24,50,51]. PDB IDs: 3UVW (**b**), 5IOK (**c**), and 5I3I (**d**). **e** Structure of the ING2 PHD finger in complex with H3K4me3[30]. PDB ID: 2G6Q. **f** A zoom-in view of the overlaid histone tail-binding sites of the ING2 PHD-H3K4me3 and MLL4 PHD6-H4K16ac complexes. **g** Alignment of amino acid sequences of the PHD fingers from human MLL4, MLL3, BPTF, and ING2. Absolutely, moderately, and weakly conserved residues are colored blue, green, and yellow, respectively. Zinc-coordinating cysteine/histidine residues are labeled

mutants were generated by site-directed mutagenesis using the Stratagene Quik-Change mutagenesis protocol, grown, and purified as WT proteins.

**NMR experiments.** NMR experiments were carried out at 298 K on Varian INOVA 600 and 900 MHz spectrometers. NMR samples for structure determination were prepared in 20 mM Tris-HCl (pH 7.0) buffer, supplemented with 100 mM NaCl, 5 mM DTT, and 8% D$_2$O. Backbone and side chain chemical shift assignments for the H4K16ac-bound MLL4-PHD6 finger were obtained by collecting and processing a set of triple resonance experiments (HNCACB, CBCA (CO)NH, CC(CO)NH, HBHA(CO)NH, HNCA, H(CCO)NH)[39] with Non Linear Sampling[40]. Chemical shift assignments were obtained for 96% of backbone amides (minus proline residues). The NMR sample contained 2.5 mM $^{13}$C/$^{15}$N-labeled MLL4-PHD6 and 7.5 mM H4K16ac (11–21) peptide (the 3-fold excess of peptide

almost saturated the 2.5 mM $^{13}$C/$^{15}$N-labeled MLL4-PHD6 sample based on $^1$H,$^{15}$N HSQC and $^1$H,$^{13}$C HSQC titration experiments with H4K16ac (11–21) peptide (Supplementary Fig. 6). Backbone chemical shift assignments for the apo-state were obtained through $^1$H,$^{15}$N HSQC titration experiments using backbone chemical shift assignments for the H4K16ac-bound MLL4 PHD6 finger. Three-dimensional (3D) $^{15}$N- and $^{13}$C-edited NOESY-HSQC spectra (mixing time of 100 ms) were collected to obtain distance restraints. Two-dimensional (2D) $^{15}$N-/$^{13}$C-filtered NOESY-HSQC and TOCSY-HSQC spectra were collected to obtain chemical shift assignments for the H4K16ac peptide. NMR experiments, including 2D NOESY, TOCSY, and reverse HSQC titration experiments, were performed on a synthetic $^{15}$N/$^{13}$C-labled H4K16ac peptide (residues 11–21, synthesized by Bio-Synthesis) to validate the assignments. 3D $^{15}$N-/$^{13}$C-filtered $^{13}$C- and $^{15}$N-edited NOESY-HSQC spectra (mixing time of 150 ms) were collected to obtain inter-molecular distance restraints.

For NMR titration experiments, $^{1}$H,$^{15}$N HSQC spectra of 0.15 mM uniformly $^{15}$N-labeled WT or mutated MLL4-PHD6 were recorded while peptides (synthesized by SynPeptide) were added stepwise. All experiments were carried out in phosphate-buffered saline buffer (pH 6.5).

**Structure determination for the MLL4-H4 complex.** Calculation of the structure of MLL4-PHD6 (aa 1503–1562) in complex with H4K16ac (11–21) peptide was carried out using interproton NOE-derived distance restraints and dihedral angle restraints. NMR spectra were processed and analyzed with NMRDraw and CcpNmr Suite[41]. The program DANGLE in CcpNmr Suite was used to predict dihedral angle $\psi$ and $\varphi$ restraints. Hydrogen bonds were derived from characteristic NOE patterns in combination with dihedral angles. The zinc-coordinating residues were initially identified through sequence alignment with other PHD fingers, and all cysteines but C1526 showed characteristic $^{13}$C$\alpha$ and $^{13}$C$\beta$ chemical shifts of zinc-ligating cysteine residues[42]. To confirm that C1526 but not the preceding H1525 residue coordinates the zinc ion, H1525 was mutated to alanine. $^{1}$H,$^{15}$N HSQC spectrum of H1525A MLL4-PHD6 reveals that this mutant is well folded and therefore H1525 is not a zinc-ligating residue (Supplementary Fig. 14). The zinc-coordinating residues were further confirmed by intramolecular distance restrains. During the calculation, sequence region $G_{13}GAK_{ac}RH_{18}$ of the H4 peptide was used with N-terminal acetyl and C-terminal N-Me amide capping groups. The structures were calculated initially with XPLOR-NIH and refined with AMBER[43,44]. Hundred structures were calculated, and the ensemble of 15 conformers with the lowest total energy was selected to represent the complex between MLL4-PHD6 and H4K16ac peptide. The quality of the structures was validated using the program PROCHECK-NMR. The percentage of residues in the most favored, additionally allowed, generously allowed, and disallowed regions is 79.9, 16.8, 2.6, and 0.7, respectively. The structural statistics is listed in Supplementary Table 1.

**Fluorescence spectroscopy.** Spectra were recorded at 25 °C on a Fluoromax-3 spectrofluorometer (HORIBA). The samples containing 1.0 μM MLL4 PHD6 domain, WT or mutants, and progressively increasing concentrations of peptide were excited at 295 nm. Experiments were performed in buffer containing 20 mM Tris (pH 7.2), 150 mM NaCl, and 1 mM DTT. Emission spectra were recorded over a range of wavelengths between 330 and 360 nm with a 0.5-nm step size and a 1-s integration time and averaged over 3 scans. The $K_d$ values were determined using a nonlinear least-squares analysis and the equation:

$$\Delta I = \Delta I_{max} \frac{\left(([L]+[P]+K_d) - \sqrt{([L]+[P]+K_d)^2 - 4[P][L]}\right)}{2[P]} \quad (1)$$

where $[L]$ is the concentration of the peptide, $[P]$ is the concentration of MLL4-PHD6, $\Delta I$ is the observed change of signal intensity, and $\Delta I_{max}$ is the difference in signal intensity of the free and bound states of MLL4-PHD6. The $K_d$ value was averaged over three separate experiments, with error calculated as the standard deviation between the runs.

**Fluorescent MST-binding assay.** The MST experiments were performed using a Monolith NT.115 instrument (Nanotemper) as described previously[26]. All experiments were performed with the purified MLL4-PHD6 domain (aa 1503–1562) in a buffer containing 20 mM HEPES (pH 7.5), 150 mM KCl, 10 mM MgCl$_2$, 10 mM β-ME, 1% sucrose, and 0.08% Tween-20. The final concentration of the N-terminal fluorescein-labeled histone H4 peptide (1–21, KE BIOCHEM) was kept at 200 nM. Dissociation constants for the interaction of MLL4-PHD6 with unlabeled peptides H4 (1–23) and H4K16ac (1–23) were measured using a displacement assay in which increasing amount of unlabeled peptides were added into a preformed PHD6:FAM-H4 complex prepared by supplementing 20 μM PHD6 into each sample. The measurements were performed at 30% LED and 40% MST power with 3 s laser-on time and 22 s off time. For all measurements, samples were loaded into standard capillaries and 900–1300 counts were obtained for the fluorescence intensity. The $K_d$ and IC$_{50}$ values were determined with the MO. Affinity Analysis software (NanoTemper Technologies GmbH), using two independent MST measurements. The $K_i$ values for unlabeled peptides with PHD6 were determined from the IC$_{50}$ values observed in the displacement assay and converted by the following equation:

$$K_i = [I]_{50} / \left( \frac{[L]_{50}}{K_d} + \frac{[P]_0}{K_d} + 1 \right) \quad (2)$$

where $[I]_{50}$ is the concentration of free unlabeled ligand at 50% binding and $[L]_{50}$ is the concentration of free labeled H4 peptide at 50% binding. The $K_d$ value is the dissociation constant of labeled H4 peptide determined in the direct binding experiment described above. Measurements for H4 unmodified and H4K16ac peptides were done in triplicates.

**Western blot and qRT-PCR.** Western blot of histone modifications using acid extracts or of nuclear proteins using nuclear extracts were done as described[45,46]. Total RNA was extracted using TRIzol (Invitrogen) and reverse transcribed using the ProtoScript II First-Strand cDNA Synthesis Kit (NEB), following the manufacturers' instructions. qRT-PCR was done using the following SYBR green primers: *Mof* (forward, 5'–TGAGATCAACCATGTGCAGAAG–3', and reverse, 5'–AGGTGAGAAATACCAGGCATC–3'). qPCR was done using Luna® Universal qPCR Master Mix according to the manufacturers' instructions (NEB). To detect truncated *Mof* transcripts after 4-OHT treatment (2 μM), RT-PCR was performed as described[10].

**Chromatin immunoprecipitation–sequencing.** In all, $2 \times 10^7$ cells were cross-linked by adding formaldehyde to a final concentration of 1.5% for 10 min. To quench crosslinking, 125 mM glycine was added. After centrifugation, pellet was resuspended in Farmham lysis buffer, and the resulting nuclear pellet was sonicated in 2 ml TE buffer supplemented with protease inhibitors to fragment size of 200–500 bp. For ChIP, 8 μg of MLL4 antibody or 4 μg of H4K16ac antibody was pre-incubated with 50 μl Dynabeads protein A (Life Technologies). Approximately 100–200 μg of chromatin was mixed with preincubated protein A-antibody complex and incubated overnight at 4 °C with rotation. ChIP samples were washed and prepared by reverse-crosslinking. DNA was purified using the QIAquick PCR Purification Kit (QIAGEN). Hundred nanograms of ChIPed DNA was used to construct sequencing library using the NEBNext Ultra II DNA Library Prep Kit for Illumina (NEB). All ChIP-Seq samples were sequenced on the Illumina HiSeq 3000.

**ChIP-Seq data analysis.** Raw sequencing data were aligned to mouse mm9 genome using bowtie2. For peak calling of MLL4 and H4K16ac enrichment, SICER algorithm was used[20]. For MLL4 peak calling, a window size of 50 bp and a gap size of 50 bp was used. For H4κ16ac, a window size of 200 bp and a gap size of 200 bp was used. Western blot data were used for normalization of H4K16ac. For GO analysis of the MLL4-associated genomic regions, GREAT version 3.0.0 was used (http://great.stanford.edu/public/html/)[47]. For enriched transcription factor motif analysis of the MLL4-binding sites, SeqPos motif tool in the Galaxy Cistrome was used (http://cistrome.org/ap/root) with default parameters[48].

**Reporting summary.** Further information on research design is available in the Nature Research Reporting Summary linked to this article.

## Data availability

Coordinates and structure factors for the MLL4-PHD6:H4K16ac complex have been deposited in the Protein Data Bank (PDB ID 6O7G). NMR spectral parameters for the MLL4-PHD6:H4K16ac complex have been deposited to Biological Magnetic Resonance Data Bank under accession number 30585. ChIP-Seq data were submitted to GEO database under accession number GSE130091. All other relevant data supporting the key findings of this study are available within the article and its Supplementary Information files or from the corresponding authors upon reasonable request. The source data underlying Figs. 1f, 3c, and 4b and Supplementary Fig. 12 are provided as a Source Data file. A reporting summary for this article is available as a Supplementary Information file.

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

## Acknowledgements

We thank Jianrong Lu for the lentiviral pLenti-CMV plasmids expressing WT or K274R mutant of MOF. This work was supported by grants from NIH GM106416, GM125195, and GM100907 to T.G.K., GM124736 to S.B.R., GM126900 to B.D.S, and DK071900 and CA129325 to R.G.R. and by the Intramural Research Program of NIDDK, NIH to K.G.

## Author contributions

Y.Z., Y.J., J.-E.L., J.W.A., L.X., M.R.H., E.M.C., K.K., B.J.K., and S.-P.W. performed experiments and, together with Y.D., R.G.R., B.D.S., S.B.R., X.S., K.G., and T.G.K., analyzed the data. Y.Z., J.-E.L., K.G., and T.G.K. wrote the manuscript with input from all authors.

## Additional information

**Competing interests:** The authors declare no competing interests.

