## [Peer Review File · Nature Communications]

Reviewers' comments:

Reviewer #1 (Remarks to the Author):

Y Zhang et al. in this manuscript described their work on identifying PHD6 finger of MLL4 as a selective reader of the H4K16ac modification. The major conclusions were supported by structural, biophysical, and genomic data. The discovery is very interesting as it expands the current understanding how acetyllysine is recognized by reader domains and is utilized for epigenetic regulation. The manuscript was well written and organized. The data were solid and clearly presented. It will be a significant publication of broad interest to the epigenetic community. Overall, the results are quite novel and valuable to publish.

Page 3, line 45-46 needs to be revised for clarity. In its current writing, it misled readers that cancer is categorized as one developmental disease.

Reviewer #2 (Remarks to the Author):

In this manuscript the authors study the interaction between the PHD6 finger domain of the MLL4 lysine methyltransferase and the N-terminal region of the histone H4. They propose that the interaction is favoured by the acetylation of the histone in Lys16, solve the solution structure of the complex and present functional correlation between the MLL4 methyltransferase and the MOF-acetyltransferase responsible of the H4 histone modification.

The topic addressed in the manuscript is important and relevant. The text is in general well-written and experiments and data are clearly presented. The authors have done an extensive work trying to corroborate their functional findings from a biophysical point of view. However, they have not convincingly demonstrated that the MLL4 PHD6 domain has a clear capacity of distinguish acetylated from non-acetylated H4 histone in position 16. Therefore, the authors should address this concern and a number of additional technical comments on the structural and biophysical characterization in a revision of the manuscript.

(1) NMR data

- It is understandable that due to clarity only small spectra regions are shown in main figures, but the complete 1H-15N HSQC spectra should appear in the Supplementary material at least for the apo form of the MLL4 PHD6 and final points of the main titration experiments (MLL4-PHD6 + H41-23 and MLL4-PHD6 + H4K16ac1-23 or H4K16ac11-21).
- The authors state that they have obtained chemical shift assignments of both apo and bound (to H4K16ac11-21) states of MLL4 PHD6, but they do not state the type of experiments acquired and the completeness of assignments. The authors do not state if the assignment of the apo form of MLL4_PHD6 was done only for backbone resonances or for the complete protein. The backbone assignment of apo and bound (to H4K16ac11-21) forms should be annotated in the complete 1H-15N HSQC spectra.
- The assignment of free MLL4-PHD6 (backbone or complete) and the assignment of the complex MLL4-PHD6 + H4K16ac11-21 should be deposited in the BNMRB database with two different entries.

(2) Selectivity of MLL4_PHD6 for the K16 acetylated version of H4 histone compared to the non-acetylated or to other acetylation patterns. The authors have done two biophysical and one biochemical experiment to address this question: titrations followed by NMR and tryptophan fluorescence spectroscopy and pull-down assays.

- The binding curves obtained based on tryptophan fluorescence show an increase of the affinity of around one order of magnitude for the acetylated histone on position K16 compared to non-acetylated and acetylated versions on K12 and K20. However, there is a lack of consistency comparing the four curves (Figure 1e and Supplementary Figure 2); for the non-acetylated, K12ac and K20ac H4 histone authors titrate MLL4_PHD6 reaching a concentration of 70 μ M of H4 peptide,

while for the H4K16ac they only reach 20 μ M. This last titration should be performed using similar concentrations (30, 50 and 70 μ M points) to discard a truncated titration and a biased fitting of the curve.

- The results of the pull-down experiments (Supplementary Figure 4) seem to agree with the fluorescence titration data. However, the intensities of the bands should be quantified, the experiments should be repeated and an adequate statistical treatment should be done (taking into account input quantities) in order to extract relevant conclusions (lines 106-108: "Pull-down experiments... ..corroborated the high selectivity of this domain toward H4K16ac and its indifference to other acetylation marks in H4 or H4K20me3").
- The conclusions that the authors draw from the NMR titrations (Figure 1c, 1d and Supplementary Figure 3) not clear: the authors state that titrations with H4 (non acetylated), H4K12ac and H4K20ac are similar while the one using H4K16ac is different. It is true that for the H4K16ac titration, some NMR signals experience a little more line broadening upon titration (assuming that the contour levels in all spectra are comparable), but the proposed "slow-to-intermediate exchange regimen" (lines 100-101) is not fully clear from this. Moreover the differences between H4K16ac titration (Figure 1d) and non-acetylated H4 (Figure 1c, left panel) are smaller than the differences between these non-acetylated H4 titration and the H4K12ac one (Supplementary Figure 3), although the authors state that these last two interactions have the same affinity. The authors should make more adequately analyse the NMR titrations beyond qualitative interpretation of a small region of the spectra: (i) they should determine the dissociation constant for each titration following the protocol that they describe on the Methods section (lines 315 to 323); this should give an idea of the affinities for each peptide. (ii) Then the authors should systematically compare the chemical shift perturbation (CSP) pattern for each titration (at least using non-acetylated and K16 acetylated peptides). For this they should plot chemical shift changes for each residue at a similar titration point (i.e. 1:5). Finally, the authors should carefully monitor the differences of binding between the H4 peptides in the region that surrounds the Lys16. If the general CSP pattern looks similar regardless of the type of H4 utilized, a similar binding mode is expected and differences should arise only in the region surrounding the acetylated lysing 16. According to their complex structure W1529, L1504 and maybe Y1514 and L1520 should be affected in a different way. One key signal to follow would be the Ne1-He1 crosspeak from the Trp1529 side chain that should appear in the 1 H- 15 N HSQC.
- In light of the lack of a solid biophysical evidence of the preference for the Lys16 acetylated version of H4 peptide, the authors should perform additional titration experiments using isothermal titration calorimetry (ITC) and/or surface plasmon resonance (SPR) to clearly demonstrate that the PHD6 domain of MLL4 acts as a H4 K16acetylation reader in vitro. In addition, these experiments would give further information on the thermodynamic (ITC) and kinetic (SPR) properties of the interaction.

(3) Structure calculation

- The coordinates of the structure should be deposited in the Protein Data Bank and the quality validation report obtained after the deposition should be provided for review together with the PDB accession code. Structural validation should be done by state-of-the-art tools, i.e. PSVS or CING.
- The authors report 30 intermolecular distance restraints to define the complex interface. This should be documented in a supplementary figure showing strips of the filtered NOESY experiments with cross peaks that were used for the distance derivation and probably referred in line 122. It is surprising that in the models (Figures 2, 3 and 7 and Supplementary Figure 5) side chains of Arg17 and His18 of the H4 peptide are shown when no intermolecular NOEs are available for these chains (as stated in lines 120-122). Thus, the conformation cannot well defined by the data.
- The authors should explain how they have implemented the Zn coordination centres in the structure calculation in the Method section. Have they done any investigation to corroborate the amino acids implicated in the Zn binding or have they relied on the sequence similarity to other PHD domains?
- The authors state in the Methods section, that they assigned the bound state of the peptide using 7.5mM of H4 peptide and 2.5mM of labelled MLL4_PHD6, having a huge contribution of free peptide. Moreover, authors do not show in the manuscript any spectra of this titration using the

shorter version of H4 (residues from 11 to 21). How complete is the assignment (at 1:3 ratio for longer H4 peptides, many peaks in MLL4_PHD6 suffer of severe line broadening effect)? Do the shortening of the peptide and the increasing of the concentration compensate this line broadening effect?

- Statements where authors explain the features of K16ac recognition (lines 127 to 130) should be expanded and supported by intermolecular NOE data (strips of filtered NOESYs showing the proximity of Leu 1504, Leu 1520 and Trp 1529 in MLL4 to the acetylated Lys16 from H4).
- In the structure description authors state that "...loops and the C-terminal α -helix showed higher flexibility..." (line 116). The apparent flexible regions in the model could just be due just to lack of NOE information. To provide experimental evidence for the proposed flexibility, the authors should provide NMR 15N T1, T2, and/or $\{^1\text{H}\}$ -15N heteronuclear NOE data.
- With mixing times of 200 to 300 ms even for a small (50 +10 residue) complex, there might be significant contributions of spin diffusion. Have the authors considered this?

(4) Mutational analysis. The authors use several approaches to validate and confirm the residues contributing to the binding of the peptide. Unfortunately none of these help on the elucidation of the preference of acetylated to non-acetylated H4 peptide:

- The work done comparing the chimeric constructs shows that the region of H4 interacting with MLL4 is the one comprising K16 and R17, but all these experiments were done in absence of acetylation. It would be interesting to explain the comparison of the H4-MLL4_PHD6 chimera (non-acetylated) with the titration done between MLL4_PHD6 and the H4K16ac peptide in left panel in Figure 3a and include an extra panel with the comparison between the chimera and the titration with the non-acetylated peptide. Which residues are differently affected by the acetylation? Does this support the structure of the complex?
- Then the authors perform eight mutations in MLL4_PHD6 and test the binding to H4. The authors state that four of these mutations lead to unfolding of the PHD domain
 - o Why do the authors test only two mutants with non-acetylated H4 and the other two with the acetylated peptide? It is more consistent to test the binding of the mutants with both peptides and check the differences; this comparison could provide important information on the acetylation recognition mechanism.
 - o Binding affinities are calculated from NMR titrations and compared to the ones obtained by fluorescence spectroscopy using wild type MLL4. This is not meaningful and values should be compared obtained with the same technique. Again a comparison between acetylated and non-acetylated H4 should be done. Figure 3e should be presented as a table with a summary of all calculated affinities.
 - o The authors state that the E1540K and E1544K mutants do not show interaction with the acetylated peptide because of the electrostatic contribution to the interaction (lines 153-155). These observations should be rationalized based on the complex structure. E1540 could be interacting with Arg17 from the peptide, but E1544 seems to be solvent exposed with no contacts to H4 peptide, why does its mutation completely abolish the interaction?
 - o How can the authors conclude (lines 155-157): "Substitution of E1516 with alanine decreased binding ~9-fold, and Y1514A mutant showed a 24-fold reduction in the binding, pointing to its role in formation or stabilization of the Lys16ac-binding groove" when they performed the titrations using the non-acetylated H4 peptide (Figure 3e)?
 - o In order to identify residues involved in the recognition of the acetylated residue, the authors should mutate amino acids showing differences in the NMR titrations with acetylated and non-acetylated H4, those that make contacts to the acetylated lysine in the structure and (i.e. L1504 or L1520) and those giving clear intermolecular NOEs.
 - o Finally, the authors should show at least a 1D spectrum of the mutants that appear to be unfolded. For example W1529A mutant was previously done in the PHD4-6 context (reference 8, Dhar, S. S. et al. 2012, Suppl. Figure 3H) and it was shown to be fully functional in H4 binding.

Minor points:

- In Supplementary Figure 1, the first panel corresponds to a titration of labelled MLL4-PHD6 with a peptide comprising the first 12 residues of H3 histone. This titration is not referred anywhere on

the text and should be either deleted or mentioned.

- In order to add clarity to the data, I recommend to use the same colour patten for the titration points in Figure 1 and Supplementary Figures 1 and 3; at least for the common ones (1:0.5, 1:1, 1:2or2.5).
- In line 108, the authors state that pull-down experiments were also done with H4K20me3 peptide, but there is no sign of these data in Supplementary Figure 4 nor elsewhere in the rest of the manuscript.
- In supplementary figure 5 the authors should show the ensemble of 15 structures containing all residues used for the pairwise r.m.s deviation calculation in Supplementary Table 1 (including His18 from the peptide).
- In supplementary Table 1 there is a typo in the legend: MLL6-PHD6 should be corrected to MLL4-PHD6.

Reviewer #3 (Remarks to the Author):

The manuscript by Zhag et al entitled: "Selective binding of the PHD6 finger of MLL4 to histone H4K16ac links MLL4 and MOF" reports that the PHD6 finger of MLL4 (MLL4-PHD6) is a reader of the common histone mark H4K16ac. The authors determine the structure of a peptide containing the H4K16ac mark with MLL4-PHD6 and determined the affinity (Kd 1 uM) of other H4 marks and mutants by HSQC titration experiments. These experiments revealed a 12-28-fold selectivity over the other acetylation marks tested. Studies in MOF deletion preadipocytes combined with ChIP-seq experiments and a catalytically inactive MOF mutant, revealed that MLL4 binding sites significantly overlapped with H4K16ac-enriched (H4K16ac+) sites in control cells and that MOF recruitment is dependent on MOF catalytic activity.

This is a very interesting study and the observation that MLL4-PHD6 as well as the closely related MLL3-PHD7 recognizes H4K16ac is unexpected. The experiments are well documented and all contain meaningful control experiments. I have therefore no major concerns recommending this manuscript for publication. However, a few minor issues should be addressed:

The author claim that MLL4-PHD6 is a selective reader of MLL4-PHD6. However, the binding site of this PHD domain contains all residues required for interaction with methylated lysines – do the authors rule out interaction with methylation marks? This would be important to investigate given the claim of acetylation specific interaction with MLL4-PHD6

MLL4 contains seven PHD6 domains – the authors should include a short section in the discussion summarizing what is known about the specificity of these domains. The ChIP experiments of the authors suggest that PHD6 is very relevant for recruitment to chromatin. Thus, do the other domains have redundant function?

We thank the Editor and Reviewers for the insightful and very constructive comments, which were helpful in revising and strengthening this manuscript.

Reviewer 1, Comment 1: ... *The discovery is very interesting as it expands the current understanding how acetyllysine is recognized by reader domains and is utilized for epigenetic regulation... It will be a significant publication of broad interest to the epigenetic community. Overall, the results are quite novel and valuable to publish.*

Page 3, line 45-46 needs to be revised for clarity. In its current writing, it misled readers that cancer is categorized as one developmental disease.

Author's response: we agree and removed the word 'developmental'.

Reviewer 2, Comment 1: ... *The topic addressed in the manuscript is important and relevant. The text is in general well-written and experiments and data are clearly presented. The authors have done an extensive work trying to corroborate their functional findings from a biophysical point of view. However, they have not convincingly demonstrated that the MLL4 PHD6 domain has a clear capacity of distinguish acetylated from non-acetylated H4 histone in position 16. Therefore, the authors should address this concern...*

Author's response: we have measured K_d s for the interactions of MLL4-PHD6 with H4 and H4K16ac by MST (Fig. 1g), which together with K_d values obtained by fluorescence confirm selectivity of MLL4 PHD6 toward H4K16ac.

Reviewer 2, Comment 2: *Additional technical comments on the structural and biophysical characterization:*

NMR data

- *the complete 1H-15N HSQC spectra should appear in the Supplementary material – the complete ^1H , ^{15}N HSQC spectra of MLL4-PHD6 + H4₁₋₂₃ and MLL4-PHD6 + H4K16ac₁₋₂₃ are shown in Suppl. Figs. 2 and 3.*

- *The authors state that they have obtained chemical shift assignments of both apo and bound (to H4K16ac11-21) states of MLL4 PHD6, but they do not state the type of experiments acquired and the completeness of assignments... The backbone assignment of apo and bound (to H4K16ac11-21) forms should be annotated in the complete 1H-15N HSQC spectra. – we have added this information to the NMR method section. The backbone assignments for the MLL4-PHD6:H4K16ac (11-21) complex and the apo-state are shown in Suppl. Fig. 7.*

- *The assignment of free MLL4-PHD6 (backbone or complete) and the assignment of the complex MLL4-PHD6 + H4K16ac11-21 should be deposited in the BMRB database with two different entries. – the complete assignment of the complex, but not of the apo-state as they were obtained through HSQC titrations, have been deposited to BMRB.*

The following sentence has been added on page 18: "NMR spectral parameters for the MLL4-PHD6:H4K16ac complex have been deposited to Biological Magnetic Resonance Data Bank under accession number 30585."

Reviewer 2, Comment 3:

Selectivity of MLL4_PHD6

- *The binding curves obtained based on tryptophan fluorescence show an increase of the affinity of around one order of magnitude for the acetylated histone on position K16 compared to non-acetylated and acetylated versions on K12 and K20. However, there is a lack of consistency comparing the four*

curves (Figure 1e and Supplementary Figure 2); for the non-acetylated, K12ac and K20ac H4 histone authors titrate MLL4_PHD6 reaching a concentration of 70 μ M of H4 peptide, while for the H4K16ac they only reach 20 μ M. This last titration should be performed using similar concentrations (30, 50 and 70 μ M points)...

– the amount of ligand added to a protein in order to reach saturation and therefore fit the data properly depends on K_d of individual reaction and is typically about 5-10-fold of the K_d value. The stronger interaction is, the less ligand is required to reach saturation. In the case of binding of H4K16ac, we used 20-fold excess of the ligand, and the binding curves clearly show that saturation has been well reached. We have split Fig. 1e into two panels to avoid confusion. In addition, we have performed one run of the titration with H4K16ac (1-23) peptide (these long acetylated peptides are expensive) using up to 70 μ M of the peptide (please see figure on the left). The K_d value remains the same.

- The results of the pull-down experiments (Supplementary Figure 4) seem to agree with the fluorescence titration data. However, the intensities of the bands should be quantified, the experiments

should be repeated and an adequate statistical treatment should be done (taking into account input quantities) in order to extract relevant conclusions... – we have replaced this non-quantitative assay with quantitative MST measurements.

- The conclusions that the authors draw from the NMR titrations (Figure 1c, 1d and Supplementary Figure 3) not clear: the authors state that titrations with H4 (non acetylated), H4K12ac and H4K20ac are similar while the one using H4K16ac is different. It is true that for the H4K16ac titration, some NMR signals experience a little more line broadening upon titration (assuming that the contour levels in all spectra are comparable), but the proposed “slow-to-intermediate exchange regimen” (lines 100-101) is not fully clear from this.... The authors should make more adequately analyse the NMR titrations beyond qualitative interpretation of a small region of the spectra... – in our hands, estimation of affinities in a low micromolar range by NMR unfortunately has not been accurate (it seems we always hit the limit around 50 μ M, no matter how strong interactions are). Therefore, we used MST assays to quantitatively measure binding affinities and confirmed K_d s measured by fluorescence. We agree and revised the phrase to: “upon titration of the H4K16ac peptide, some resonances of MLL4-PHD6 exhibited more pronounced line broadening...”

- In light of the lack of a solid biophysical evidence of the preference for the Lys16 acetylated version of H4 peptide, the authors should perform additional titration experiments... to clearly demonstrate that the PHD6 domain of MLL4 acts as a H4 K16acetylation reader in vitro. – as suggested, we have measured binding affinities by MST (Fig. 1g). The MST data correlate well with the tryptophan fluorescence data and confirm the selectivity of MLL4-PHD6 for H4K16ac.

Reviewer 2, Comment 4:

Structure calculation

- The coordinates of the structure should be deposited in the Protein Data Bank and the quality validation report obtained after the deposition should be provided for review. – the structure has been deposited (PDB ID 6O7G), and the final validation report is included.

- The authors report 30 intermolecular distance restraints to define the complex interface. This should be documented in a supplementary figure showing strips of the filtered NOESY experiments with cross

peaks that were used for the distance derivation and probably referred in line 122. It is surprising that in the models (Figures 2, 3 and 7 and Supplementary Figure 5) side chains of Arg17 and His18 of the H4 peptide are shown when no intermolecular NOEs are available for these chains (as stated in lines 120-122). Thus, the conformation cannot well defined by the data. – Suppl. Fig. 8 has been added to show intermolecular NOEs.

The sentence (page 6) has been revised to: "...Gly14, Ala15, Lys16ac and Arg17 of H4 and residues L1519, L1520, I1521, Q1522, W1529, E1540, V1543 and A1547 of MLL4-PHD6." His18 of H4 has only one weak NOE, and we agree, it's conformation cannot be well defined. The side chain of His18 has been removed in Figures.

- The authors should explain how they have implemented the Zn coordination centres in the structure calculation in the Method section. Have they done any investigation to corroborate the amino acids implicated in the Zn binding or have they relied on the sequence similarity to other PHD domains? – the method section (page 15) has been expanded to clarify this: "The zinc coordinating residues were initially identified through sequence alignment with other PHD fingers, and all cysteines but C1526 showed characteristic $^{13}\text{C}\alpha$ and $^{13}\text{C}\beta$ chemical shifts of zinc-ligating cysteine residues⁴³. To confirm that C1526 but not the preceding H1525 residue coordinates the zinc ion, H1525 was mutated to alanine. ^1H , ^{15}N HSQC spectrum of H1525A MLL4-PHD6 reveals that this mutant is well folded and therefore H1525 is not a zinc binding residue (Suppl. Fig. 14). The zinc-coordinating residues were further confirmed by intramolecular distance restraints."

- The authors state in the Methods section, that they assigned the bound state of the peptide using 7.5mM of H4 peptide and 2.5mM of labelled MLL4_PHD6, having a huge contribution of free peptide. Moreover, authors do not show in the manuscript any spectra of this titration using the shorter version of H4 (residues from 11 to 21). How complete is the assignment (at 1:3 ratio for longer H4 peptides, many peaks in MLL4_PHD6 suffer of severe line broadening effect)? Do the shortening of the peptide and the increasing of the concentration compensate this line broadening effect? – that is correct, increasing concentrations has compensated line broadening. The NMR titration with the shorter version of H4K16ac (11-21) is shown in Suppl. Fig. 6.

- Statements where authors explain the features of K16ac recognition (lines 127 to 130) should be expanded and supported by intermolecular NOE data (strips of filtered NOESYs showing the proximity of Leu 1504, Leu 1520 and Trp 1529 in MLL4 to the acetylated Lys16 from H4). – intermolecular NOEs, including between K16ac and W1529, are shown in Suppl. Fig. 8.

- In the structure description authors state that "...loops and the C-terminal α -helix showed higher flexibility..." (line 116). The apparent flexible regions in the model could just be due just to lack of NOE information. To provide experimental evidence for the proposed flexibility, the authors should provide NMR ^{15}N T1, T2, and/or $\{^1\text{H}\}$ - ^{15}N heteronuclear NOE data. – we agree and deleted this phrase.

- With mixing times of 200 to 300 ms even for a small (50 +10 residue) complex, there might be significant contributions of spin diffusion. Have the authors considered this? – we recorded and compared crosspeak intensities in 2D of 3D filtered NOESY spectra with 150 ms, 200 ms and 300 ms mixing times and used 150 ms to run 3D filtered experiments.

Reviewer 2, Comment 5:

Mutational analysis

- ...It would be interesting to explain the comparison of the H4-MLL4_PHD6 chimera (non-acetylated) with the titration done between MLL4_PHD6 and the H4K16ac peptide in left panel in Figure 3a and include an extra panel with the comparison between the chimera and the titration with the non-acetylated peptide. Which residues are differently affected by the acetylation? Does this support the

structure of the complex? – we have added two paragraphs on pages 7-8 that describe chemical shift differences in the H4K16ac-bound and H4-bound protein. We also added spectrum of MLL4-PHD6:H4 to Fig. 3a and included Suppl. Fig. 10 to show the differences. This additional analysis fully supports the structure of the complex.

• *...Why do the authors test only two mutants with non-acetylated H4 and the other two with the acetylated peptide... Binding affinities are calculated from NMR titrations and compared to the ones obtained by fluorescence spectroscopy using wild type MLL4. This is not meaningful and values should be compared obtained with the same technique...* – we agree, the Y1514A and E1516A mutants were made and tested at the very beginning of this study, before we determined the structure of the complex and defined the histone binding site and before we found that acetylation enhances the binding. The K_d values of these mutants for non-acetylated H4, calculated by NMR, have been removed.

o The authors state that the E1540K and E1544K mutants do not show interaction with the acetylated peptide because of the electrostatic contribution to the interaction (lines 153-155). These observations should be rationalized based on the complex structure. E1540 could be interacting with Arg17 from the peptide, but E1544 seems to be solvent exposed with no contacts to H4 peptide, why does its mutation completely abolish the interaction? – the second paragraph on page 8 has been revised accordingly.

o *In order to identify residues involved in the recognition of the acetylated residue, the authors should mutate amino acids showing differences in the NMR titrations with acetylated and non-acetylated H4, those that make contacts to the acetylated lysine in the structure and (i.e. L1504 or L1520) and those giving clear intermolecular NOEs.* – the structure of the complex shows that the acetyl moiety of K16ac is bound in a hydrophobic groove formed by the side chains of W1529 and L1504. We mutated both and examined binding of L1504E to H4 and H4K16ac by fluorescence (W1529A was unfolded). The text on page 7 (last paragraph) and Fig. 3c have been added.

o *Finally, the authors should show at least a 1D spectrum of the mutants that appear to be unfolded. For example W1529A mutant was previously done in the PHD4-6 context (reference 8, Dhar, S. S. et al. 2012, Suppl. Figure 3H) and it was shown to be fully functional in H4 binding.* – ^1H , ^{15}N HSQC spectra of the mutants, including unfolded W1529A, are shown in Suppl. Fig. 11.

Reviewer 2, Comment 6: Minor points:

- *In Supplementary Figure 1, the first panel corresponds to a titration of labelled MLL4-PHD6 with a peptide comprising the first 12 residues of H3 histone. This titration is not referred anywhere on the text and should be either deleted or mentioned.* – reference to this figure has been added to Fig. 1 legend.
- *In order to add clarity to the data, I recommend to use the same colour patten for the titration points in Figure 1 and Supplementary Figures 1 and 3; at least for the common ones (1:0.5, 1:1, 1:2or2.5).* – the figures have been re-colored as suggested.
- *In line 108, the authors state that pull-down experiments were also done with H4K20me3 peptide, but there is no sign of these data in Supplementary Figure 4 nor elsewhere in the rest of the manuscript.* – this figure has been removed.
- *In supplementary figure 5 the authors should show the ensemble of 15 structures containing all residues used for the pairwise r.m.s deviation calculation in Supplementary Table 1 (including His18 from the peptide).* – His18 was not included for r.m.s.d. calculation and therefore is not shown.
- *In supplementary Table 1 there is a typo in the legend: MLL6-PHD6 should be corrected to MLL4-PHD6.* – corrected, thank you.

Reviewer 3, Comment 1:

This is a very interesting study and the observation that MLL4-PHD6 as well as the closely related MLL3-PHD7 recognizes H4K16ac is unexpected. The experiments are well documented and all contain meaningful control experiments. I have therefore no major concerns recommending this manuscript for publication. However, a few minor issues should be addressed: The author claim that MLL4-PHD6 is a selective reader of MLL4-PHD6. However, the binding site of this PHD domain contains all residues required for interaction with methylated lysines – do the authors rule out interaction with methylation marks?...

Author's response: all our attempts to test binding of MLL4-PHD6 to PTM-containing, including methylated, histone peptides in arrays performed independently multiple times in two labs (X.S. and S.R.) were unfortunately unsuccessful (one example is shown in figure below). In the complex, acetylated K16 is bound in a narrow groove, formed by W1529 and L1504, which may not be wide enough to accommodate trimethylated lysine. We have added the discussion on differences between recognition of H3K4me3 and H4K16ac by PHD fingers on page 11, last paragraph.

Reviewer 3, Comment 2:

MLL4 contains seven PHD6 domains – the authors should include a short section in the discussion summarizing what is known about the specificity of these domains. The ChIP experiments of the authors suggest that PHD6 is very relevant for recruitment to chromatin. Thus, do the other domains have redundant function?

Author's response: neither functions nor structures of PHD fingers of MLL4 are known. We have added the following sentence in discussion (page 12): “Alignment of amino acid sequences of seven PHD fingers from MLL4 and eight PHD fingers from MLL3 reveals that the H4K16ac-binding site residues of MLL4-PHD6 are conserved in MLL3-PHD7 but not in other PHD fingers of MLL4 and MLL3 for which biological roles remain unidentified (Fig. 7g).”

REVIEWERS' COMMENTS:

Reviewer #2 (Remarks to the Author):

The authors have addressed my previous concerns:

To address the selectivity for the acetylated histone compared to the non-acetylated one, they now present two MST titrations with both peptides and they reproduce the affinities they observe by fluorescence. In addition, they show the complete fluorescence titration for the acetylated peptide (they were showing only upto 20uM excess and utp 70uM for the rest). They have also removed the pull-down data instead doing a proper quantification.

To provide a proper documentation of the NMR data they provide the complete spectra together with the amide assignments, and have updated the material and methods and also give pdb and BMRB codes (for the complex). The validation report for the structure looks ok.

The are now comparing affinities for wildtype and mutants using the same technique and have better explained the lack of binding for some of them.

I recommend publication.

Reviewer #3 (Remarks to the Author):

I had only minor concerns when reviewing the previous version of this manuscript. The authors clarified now these issues and I have no further concerns regarding this paper. I congratulate the authors to this interesting study and support publication of the current version.